# Estimating oxygen uptake in simulated team sports using machine learning models and wearable sensor data: A pilot study

Dermot Sheridan[1]*, Arne Jaspers[2,3], Dinh Viet Cuong[1,4], Tim Op De Beéck[5], Niall M. Moyna[6], Toon T. de Beukelaar[3,7], Mark Roantree[1,4]

1 School of Computing, Dublin City University, Dublin, Ireland, 2 Research Group for Musculoskeletal Rehabilitation, Department of Rehabilitation Science, KU Leuven, Leuven, Belgium, 3 KU Leuven Institute of Sports Sciences, KU Leuven, Leuven, Belgium, 4 Insight Centre for Data Analytics, Dublin City University, Dublin, Ireland, 5 Runeasi, Leuven, Belgium, 6 School of Health and Human Performance, Dublin City University, Dublin, Ireland, 7 Movement Control and Neuroplasticity Research Group, Department of Movement Sciences, KU Leuven, Leuven, Belgium

* dermot.sheridan36@mail.dcu.ie

**Data availability statement:** The data can be accessed on Zenodo at the following DOI: 10.5281/zenodo.14609092.

## Abstract

Accurate assessment of training status in team sports is crucial for optimising performance and reducing injury risk. This pilot study investigates the feasibility of using machine learning (ML) models to estimate oxygen uptake ($VO_2$) with wearable sensors during team sports activities. Six healthy male team sports athletes participated in the study. Data were collected using inertial measurement units (IMU), heart rate monitors, and breathing rate sensors during incremental fitness tests. The performance of different ML models, including multiple linear regression (MLR), XGBoost, and deep learning models (LSTM, CNN, MLP), was compared using raw and engineered features from IMU data. Results indicate that while LSTM models with raw IMU data provided the most accurate predictions (RMSE: 4.976, MAE: 3.698 mL $\cdot$ kg$^{-1}$ $\cdot$ min$^{-1}$), MLR models remained competitive, especially with engineered features. Multi-sensor configurations, particularly those including sensors on the torso and limbs, enhanced prediction accuracy. The findings demonstrate the potential of ML models to monitor $VO_2$ noninvasively in real-time, offering valuable insights into the internal physiological demand during team sports activities.

## Introduction

In team sports, including the various football codes, accurately assessing the players' training status is of significant interest to coaches and sports scientists [1]. Systematic monitoring of the training process allows the evaluation of athletes' physical output and internal physiological responses [2]. Various tracking technologies, including Global Navigation Satellite System (GNNS) and accelerometry, are available for monitoring external load [3], while monitoring heart rate (HR) and rating of perceived exertion (RPE) are standard methods for assessing

**Funding:** This work was conducted with the financial support of Science Foundation Ireland under grant numbers SFI/12/RC/2289_P2 and SFI/18/ CRT/6223. SFI, Insight Research Centre for Data Analytics, URL: https://www.sfi.ie/sfi-research-centres/insight, SFI/12/RC/2289_P2, Mark Roantree SFI, Center for Research Training in Artificial Intelligence, URL: https://www.crt-ai.ie, SFI/18/ CRT/6223, Dermot Sheridan. The funders had no role in study design, data collection and analysis, decision to publish, or preparation of the manuscript.

**Competing interests:** The authors have declared that no competing interests exist.

an athlete's internal load [4,5]. Understanding the relationship between internal and external loads is crucial as it provides insights into an athlete's adaptations to training, indicating changes in fitness or fatigue state [6]. It is highly beneficial to connect external training load measures to relevant outcomes for effective training [1].

Maximal oxygen uptake ($VO_2$max) is a traditional indicator of an athlete's aerobic power, which is crucial for sustaining high-intensity efforts over prolonged periods in team sports like soccer [7]. Studies have shown that players with higher $VO_2$max values tend to cover more distance during games, a critical factor for running-based team sports [8]. Therefore, monitoring physiological parameters is crucial; $VO_2$max and oxygen consumption at anaerobic threshold parameters are essential for assessing the metabolic demands of different field roles in team sports, including soccer [9]. However, due to scheduling, such testing, like spiroergometry or cardiopulmonary fitness assessments (e.g., $VO_2$max testing), often proves impractical during competitive periods [10]. Consequently, finding a method to monitor these changes unobtrusively throughout the season is key to accurately assessing training loads and tracking athletes' physical fitness [11]. Such an approach would negate the need for frequent, invasive testing procedures like $VO_2$max tests, traditionally used to gauge adaptations over time.

A recent systematic review of the relationship between external, wearable sensor-based, and internal parameters emphasises two challenges. The first is whether we can capture and quantify the complex loading of team sports [12]. Running-based team sports are intermittent sports, consisting of hundreds of brief and very intense actions, such as jumps, tackles, changes of direction, accelerations, and decelerations [13]. Thus, more specific measures and devices are needed to identify loading in sports. IMU represent a valuable integration of sensor technologies, typically including 3D accelerometers, 3D gyroscopes, and 3D magnetometers in a single device. IMU data (100 Hz), as captured by current wearable devices in team sports, provides a sensitive measure of high-intensity actions [14]. This data has been used to create custom accelerometer metrics, which quantify three-dimensional movements and have been used to estimate $VO_2$ with varying accuracy for different physical activities [15,16]. These calculations reduce the 3D raw accelerometer data to a 1D vector, which is more convenient to handle but could result in losing value information. Additionally, the gyroscope data may offer a way to capture more information about movement during team sports. When combined with accelerometer data, it has been shown to enhance fatigue detection in runners [17].

The second challenge is whether we can accurately model the relationship between the sensors-based measure of external load and the athlete's individual internal response [12]. Traditional statistical models are limited in modelling human physiological responses as they rely heavily on significant predictors and struggle with non-linear relationships and complex data. ML provides advantages, such as interpreting complex and non-linear patterns essential for accurate predictions and understanding of physiological responses [18]. Different from traditional ML, deep neural networks can process raw data directly and autonomously learn to identify complex, hidden features, thereby eliminating the need for manual feature extraction [19]. Notably, in the development of models for action pattern recognition, the prevalent deep models include Convolutional Neural Networks (CNN), Long-Short-Term Memory Networks (LSTM), along with their hybrid forms (Chang et al. 2023). Temporal convolutional neural networks using cardiorespiratory biosignals and power as input have been utilised to accurately predict $VO_2$ responses to varying exercise intensities, leveraging past data to forecast future cardiorespiratory dynamics [20]. Such models have demonstrated capability in estimating slower $VO_2$ kinetics with increasing exercise intensity, facilitating nonintrusive monitoring across different exertion levels [21]. Using a non-linear ML model with heart rate,

respiration rate, and acceleration data from medical-grade wearables as input significantly reduced $VO_2$ estimation errors during the Bruce treadmill test, a progressively intense workout. This method has shown superior performance compared to previous heart rate-based estimations [22].

Further, integrating motion data from IMU, GNNS, and a heart rate sensor has markedly enhanced the accuracy of $VO_2$ estimation during outdoor running and walking, underlining the potential of neural networks for real-time assessments [23]. A personalised LSTM neural network model, trained on heart rate, mechanical power output, cadence, and respiratory rate, estimated individual $VO_2$ responses with high predictive accuracy across a range of cycling intensities [24]. This suggests a pathway for creating personalised models that can account for individual variations to improve the estimation of $VO_2$, enabling precise monitoring during games and training. Accurately estimating $VO_2$ for each player is an essential step in determining the contributions of the aerobic energy system to activities. Techniques like the excess post-exercise oxygen consumption plus delta lactate method (EPOC [La-]) and the accumulated oxygen deficit method (O2deficit) can provide insights into the anaerobic contribution, with anaerobic contributions accounting for a significant portion during high-intensity intermittent activities [25]. These advancements allow for a more accurate determination of the internal response, which helps us to evaluate the athlete's training status.

This study compared the accuracy of various deep learning ML models in estimating individuals' $VO_2$ from wearable sensor data during outdoor jogging and simulated team sports activities. We compared the prediction accuracy for the ML model using different IMU data representations of raw and engineered features. Finally, we analysed various combinations of body-worn accelerometers to evaluate their impact on $VO_2$ prediction. This comparative study seeks to determine the most effective data input method for ML models to estimate $VO_2$ during the high-intensity actions typical in team sports.

## Materials and methods

### Participants

A total of six healthy male team sports athletes (height: 182.55 ± 3.64 cm; mass: 79.62 ± 11.26 kg; $VO_2$max: 56.78 ± 3.83 mL·kg$^{-1}$·min$^{-1}$) participated in the study. See Table 1 for detailed participant baseline characteristics. Participants were selected based on the following inclusion criteria: a minimum of four training sessions per week over two years, with at least three sessions being pitch-based training or games. Eligibility also required no self-reported history of metabolic, neurological, pulmonary, or cardiovascular diseases and no symptoms of lower extremity injuries for at least six months prior to the study. All participants provided written informed consent in accordance with the Declaration of Helsinki. The study was approved by the local ethics committee of Dublin City University (DCUREC/2021/256).

### Protocol/data acquisition

Each athlete participated in two sessions at Dublin City University (DCU), spaced at least 48 hours apart. The first laboratory visit comprised three phases to assess $VO_2$: a resting phase, a sub-maximal exercise protocol, and a maximal graded exercise test (GXT). During the resting phase, $VO_2$ and respiratory frequency were averaged over five minutes to establish baseline metabolic rates [24]. The sub-maximal trial started at 9 km/h, increasing by 1 km/h every six minutes until blood lactate levels reached ≥ 3 mmol/L, with intermittent 1-minute rest periods for lactate sampling [26]. The treadmill gradient remained fixed at 1% to simulate the energetic cost of outdoor running [27]. Following a 5-minute rest, the maximal

**Table 1. Participant VO$_2$max and Resting VO$_2$.**

| Subject ID | VO$_2$max (mL·kg$^{-1}$·min$^{-1}$) | VO$_2$ Rest (mL·kg$^{-1}$·min$^{-1}$) |
|---|---|---|
| Subject 2 | 49.3 | 6.48 |
| Subject 3 | 59.5 | 6.37 |
| Subject 4 | 59.2 | 5.74 |
| Subject 5 | 57.0 | 6.71 |
| Subject 7 | 56.9 | 4.97 |
| Subject 8 | 58.8 | 8.76 |

Notes: **VO$_2$ max**: Maximum oxygen uptake, a measure of aerobic capacity. **VO$_2$ rest**: Resting oxygen uptake, the oxygen consumption at rest.

ramp incremental test commenced, starting at a speed 1 km/h below the final sub-maximal speed and increasing by 1 km/h each minute until reaching 16 km/h, followed by incremental increases in slope by 1% each minute until voluntary exhaustion. All tests were conducted under similar conditions (20–21 °C).

The second visit involved on-field tests on a synthetic pitch, incorporating a steady-state jog and an intermittent team sport simulated circuit developed from existing protocols [28]. Each circuit included three counter-movement jumps, an eight-meter jog, an eight-meter change of direction (COD) agility section, two jumps for distance, a 10 m sprint, seven meters of walking, and a tackle bag to be hit with force. These activities are designed to reflect the dynamic nature of team sports and lasted approximately 45 seconds, followed by 15 seconds of rest, repeated five times.

## Sensor measurements

This section describes the data measured during the laboratory and field visits and the features used for the dynamic oxygen prediction models. Fig 1 shows the measurement setup of wearable sensors worn by the athlete during the protocol.

**Oxygen Uptake (VO$_2$):** Pulmonary gas exchange data were captured using a Cosmed K5 breath-by-breath metabolic analyser, calibrated with a specific gas mixture and a flow meter before each session. VO$_2$ values were normalised by body mass, providing a detailed measure of aerobic capacity for each breath [29].

**Heart Rate (HR) & Breathing Rate (BR):** HR and BR were monitored using Zephyr Bio Harness 3.0, a validated tool for physiological monitoring in sports settings [30]. The device was worn on the chest to ensure accurate measurement of cardiac and respiratory parameters.

**Inertial Measurement Units (IMU):** IMU measured linear acceleration, angular velocity, and magnetic field variations at five body locations: the lower back, both tibiae, and both wrists. These placements were chosen to capture whole-body movement dynamics, including limb-specific and core-generated movements relevant to team sports activities. Sensors were secured with adjustable straps and calibrated before each session to ensure alignment with anatomical landmarks and reduce signal noise. Data were sampled at 250 Hz, providing high-resolution motion data.

The IMU collected multidimensional signals, including: - Linear acceleration: Used to identify movement intensity and transitions between activity states. - Angular velocity: Captured rotational dynamics, particularly during changes in direction - Magnetic field variations: Used for orientation tracking to complement acceleration and angular velocity data.

**Calibration procedure:** All wearable sensors underwent a multi-step calibration process prior to data collection: 1. Static Calibration: The sensors were placed on a stable surface

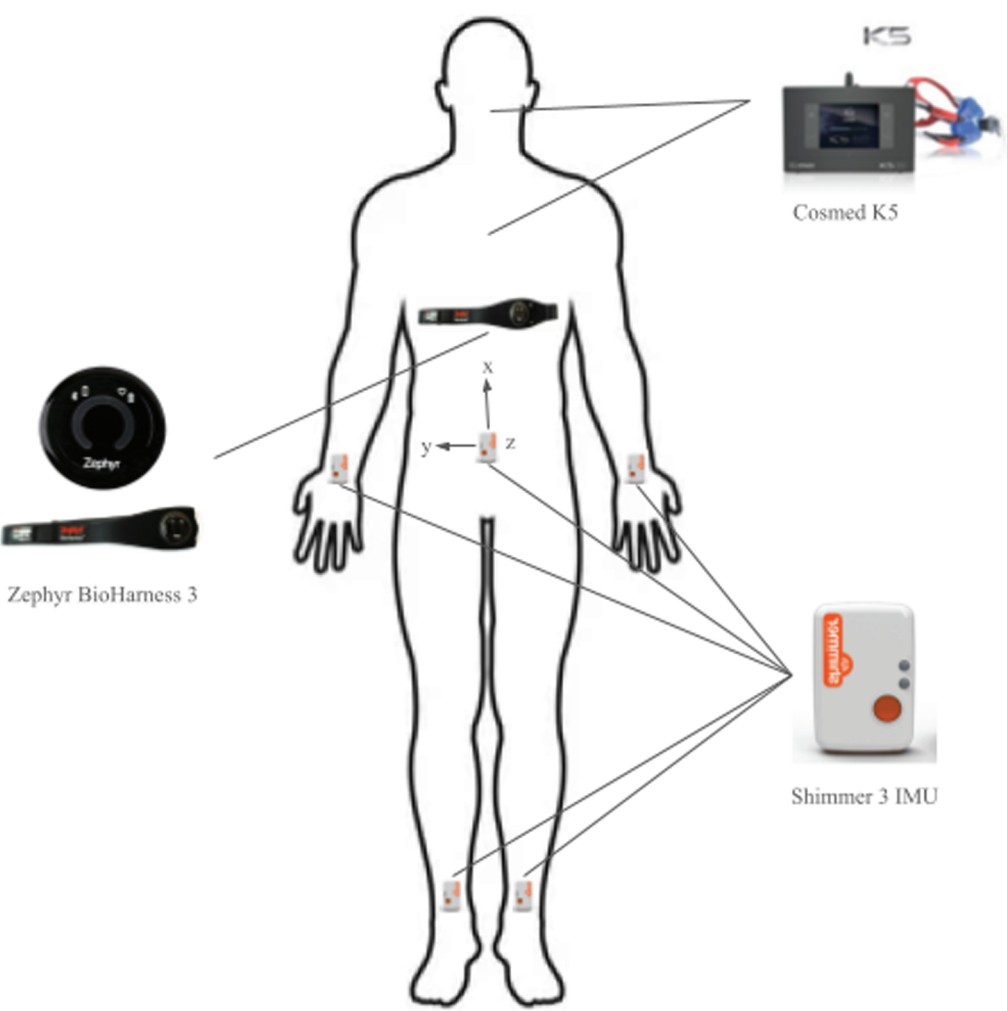

**Fig 1. Measurement setup: The Cosmed K5 portable Metabolic (the gas analyser is worn on the back, and the face mask covers the mouth and nose), the Bio Harness 3 HR and BR device is worn under the shirt, and the Shimmer 3 IMU sensors are attached at five locations on the athlete's bodies.**

to establish a baseline for signal offset correction 2. Dynamic Calibration: Participants performed a series of controlled movements (for example, walking, jogging, and arm swings) to synchronise signals across all devices. 3. Signal Quality Check: Data were inspected in real time to confirm synchronisation and detect potential artifacts.

This setup ensured accurate and reliable data collection across all test conditions, facilitating detailed feature extraction for oxygen uptake prediction.

## Pre-processing

For each treadmill test, the athletes $VO_2$max was calculated as the maximum value of the rolling average of the $VO_2$ signal with a window length of 30 seconds. Recommendations for exercise physiologists to adopt these data processing strategies to reduce variability in $VO_2$ measurements have been published. A 15-breath average can correct a residual error in $VO_2$ datasets to within 10% of the raw variability [31]. We adapted to smoothed $VO_2$ with a 31-point moving average window to reduce interference noise [22]. We repeated the calculation

of the maximum value from the 31-point moving average window of the $VO_2$ signal for comparison. The treadmill speed (km/h) for each stage of the sub-maximal and maximal test was added to the $VO_2$ data for visit 1; for the outdoor test, the GNSS speed (km/h) recorded on the Cosmed K5 device was used to determine the Speed of outdoor running. The Activity Four class labels (Resting, Treadmill Running, Outdoor Running, Simulated Team Sports Circuit) to describe the movement during the protocol were engineered and added to the $VO_2$ data. The subject's physical characteristics, age (yrs), height (cm), weight (kg), resting oxygen uptake ($VO_2$rest), and $VO_2$max were included as features (Table 1). The Zephyr data (1Hz) was directly merged into the breath-by-breath $VO_2$ data. The 5 Shimmer IMU data files were merged into one single file. To achieve this, the data was resampled from 250Hz to 125Hz to facilitate matching times. This raw IMU data was merged with the breath-by-breath $VO_2$ data to preserve its frequency. The IMU data is marked by windows of each breath recorded in the $VO_2$ data, and an example of the data can be seen in Fig 2. Due to issues with two sensors (right arm and left leg) during different sessions that failed to record during the experiment, these two of the five IMU sensors were removed from the final data. Data only from the torso, right tibia and left wrist were used. The Magnetometer data was excluded from the analysis. This forms the RAW dataset.

## Data representation

To investigate the impact of dataset representation on estimating oxygen uptake during team sports activity, the time series data were represented in a different format by transforming the raw data. The 6-axis data of the IMU sensor that consists of 3-axis acceleration and 3-axis angular velocity was engineered into axis-specific mean amplitude deviation (MAD) values, and their resultant MADxyz were determined as follows:

$$\text{MAD}_{xyz} \quad = \quad \sqrt{\frac{1}{N}\sum_{i=1}^{N}(x_i - \overline{X})^2 + \frac{1}{N}\sum_{i=1}^{N}(y_i - \overline{Y})^2 + \frac{1}{N}\sum_{i=1}^{N}(z_i - \overline{Z})^2} \qquad (1)$$

The calculations were computed on the window of IMU data between the breaths, and the dataset was compressed into a single row for each breath. The MADxyz is sensitive to changes in the axis's inclination angle and movement, making its magnitude always greater than or equal to MAD [15]. The same procedure was used to process each IMU sensor. In this way, two MADxyz (Accel xyz, Gyro xyz) individual features were obtained from one IMU, and six features were calculated for three IMUs. Taken together, this forms the engineered features dataset for the experiments.

## Data structuring

The input data structure can significantly influence on the deep learning performance results. Four input data structures of 1, 3, 5, and 7 breath windows were studied here for better input adaptation.

## Machine learning approach

A supervised machine learning regression approach was utilised, employing a modified Leave-One-Subject-Out (LOSO) cross-validation strategy. This approach was selected to mimic real-world scenarios where models are deployed to predict outcomes for new, unseen individuals. By leaving one subject out for testing while training the model on data from all

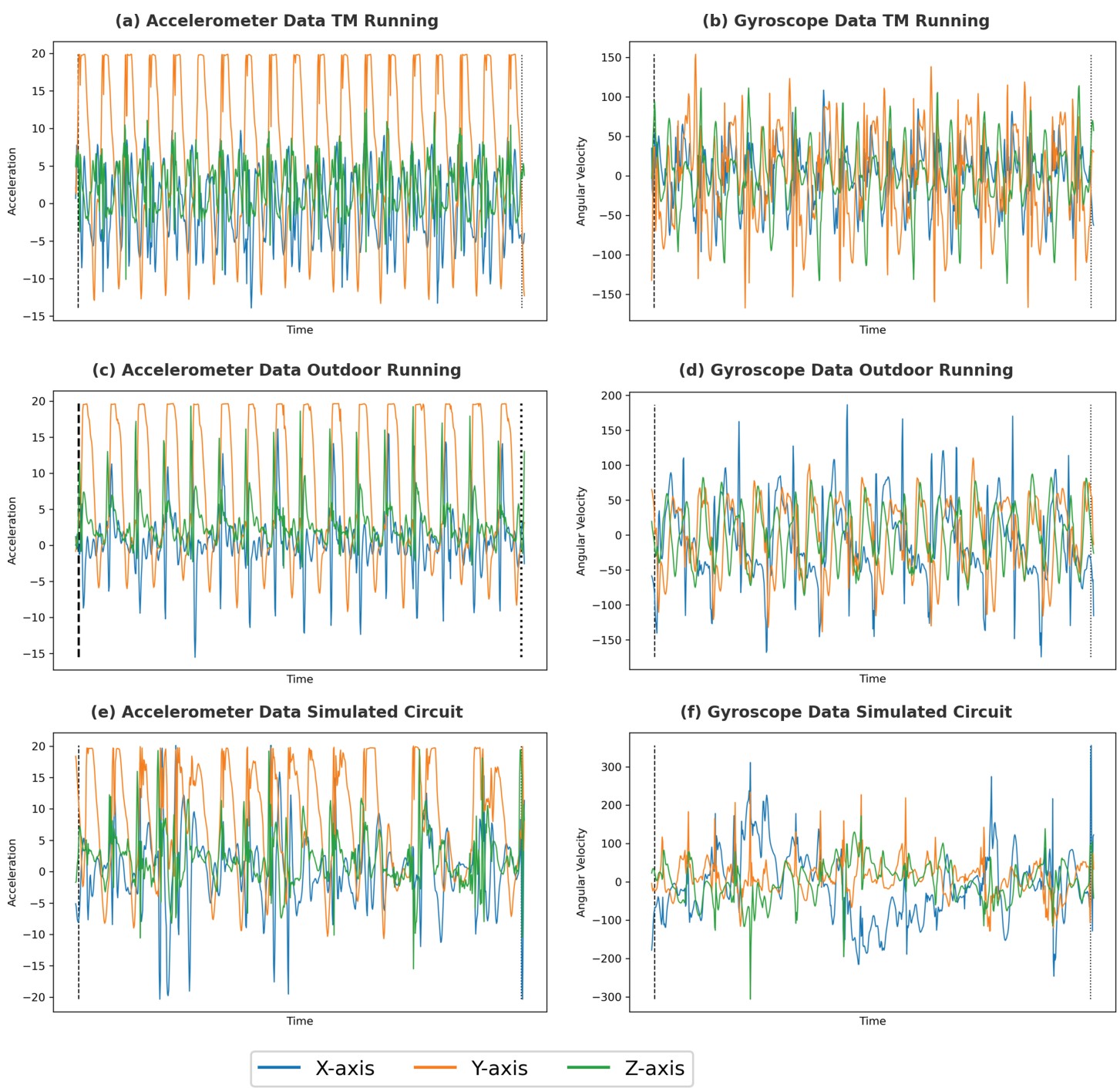

**Fig 2. Examples of IMU raw data from the Accelerometer and Gyroscope over a 3-breath window during three different activities: Treadmill Running (TM), Outdoor Running, and a Simulated Circuit.**

other subjects, LOSO cross-validation evaluates the model's generalisation capability across individuals.

Our approach is inline by methodologies used in similar studies, such as Zignoli et al. (2020), which split trials into training and testing sets to evaluate model generalisability.

In our study, data from the first visit of the test subject was included in the training phase, allowing the model to capture intra-individual variability before testing on data from the second visit. This modification ensures that the model is evaluated on entirely unseen session-level data while benefiting from information about the individual's general fitness profile, simulating practical applications in sports monitoring.

A portion of the training data was set aside as a validation set to ensure the model's reliability and accuracy. The validation set was used to select hyperparameters by minimising the root mean square error (RMSE). A grid search technique was employed to systematically identify the optimal hyperparameters, such as the number of layers and neurons, that achieved the lowest RMSE. This method ensures an unbiased and exhaustive search across the hyperparameter space, improving model performance and robustness. The same hyperparameter selection process was applied uniformly across all models to maintain consistency in model evaluation.

**Linear regression.** Linear regression [32] is a linear method for modeling the relationship between dependent and independent variables by fitting a linear function to observed data. Its simplicity and interpretability make it a common starting point for evaluating more complex models. The coefficients of the function are derived from minimising the difference between the predicted values (outputs of the linear function) and actual data points. In this study, principal component analysis (PCA) was applied to transform the inputs before being forwarded to the linear regression model. Although PCA was not strictly necessary given the limited number of features, it was included to ensure consistency with standard preprocessing practices and to represent the input data in a compact, orthogonal space. Experiments revealed that retaining all six principal components yielded the best test scores, as dimensionality reduction did not improve performance. This preprocessing step concentrated information in the first few components, which, in theory, could facilitate easier learning for the model This approach maintains the computational efficiency and interpretability of the model while allowing for a direct comparison with more complex models to estimate the breath-by-breath $VO_2$.

**Non-linear regression model.** XGBoost (Extreme Gradient Boosting): XGBoost is a powerful machine learning technique, its ability to handle non-linear relationships and its reputation for high performance and speed in both regression and classification problems, building on the Gradient Boosting Decision Trees (GBDT) framework. It optimises model performance and computational efficiency-specific advantages such as handling missing data and regularisation features to prevent overfitting and scalability for large datasets. In the XGBoost regression model, predictions result from summing the outputs of K decision trees, allowing for complex nonlinear relationships [33].

**Architecture details:** Various configurations were tested, including different numbers of trees (Number of Trees: 500, 1000, 3000, 5000) and depths (Maximum Depth: 3, 5, 7, 10).

## Deep learning models

Deep learning models are considered as their architectures have demonstrated the ability to capture complex patterns and temporal dependencies in sequential data, which is critical for estimating $VO_2$ in dynamic sports performance [19]. The impact of different deep learning neural network models was analysed to predict breath-by-breath $VO_2$ using raw time series data as input.

**Multi-Layer Perceptron (MLP).** MLP networks are adept at handling non-linearity, internal randomness, and long-term unpredictability in time series data; they transform the high-dimensional input data into a manageable latent space to make accurate predictions [34].

**Architecture details:** The activation function is ReLU; there is no dropout, and L2-regularisation is used with the weight decay 1e-4 to reduce overfitting. Various configurations test different depths (from one to four layers) and widths (32 or 64 neurons per layer), examining how each configuration affects the model's performance.

**Long Short-Term Memory (LSTM).** The LSTM model is well-suited for time series predictions due to its ability to remember patterns based on previous timestep states, making it effective in capturing long-term dependencies within sequential data [35]. We adopt a bidirectional LSTM version, which processes its inputs in a bidirectional manner along the breath dimension to capture contextual information from past and future breaths, enhancing predictive accuracy.

**Architecture details:** In LSTM, we consider breath a time step, so we have a fixed length (7, 5, 3, 1 breath); no padding is needed. There is no dropout; we use regularisation with the weight decay 1e-4 to reduce overfitting. We experiment with various configurations, ranging from 1 to 4 hidden layers, each with 32 or 64 units.

**Convolutional Neural Network (CNN).** A CNN is well-suited for time series predictions due to its effectiveness in extracting local temporal features from sequential data [36]. The model uses a series of convolutional layers; each CNN layer has kernels to slide the breath dimension, computing and extracting temporal features at every single timestep. Like the LSTM, the CNN focuses on the output from the middle breath for final processing, ensuring relevance to the temporal centre of the data.

**Architecture details:** A fully connected layer initially processes each breath and generates breath-level latent vectors prepared for subsequent 1D convolution. The 1D convolution is performed along the breath dimension, utilising 'same' padding to maintain the original input shape by adding zero padding at the edges. A stride of 1 is used, ensuring the convolution operation moves along the breath dimension one step at a time. The CNN configurations vary in depth, ranging from 3 to 5 convolutional layers and in the number of kernels, using either 32 or 64 kernels per layer. This variability allows the model to learn features at different levels of abstraction. A consistent kernel size of 3 is applied across all convolutional layers, effectively capturing local temporal patterns while maintaining broader contextual information.

**Sensors configurations.** The study also investigates the impact of various sensor placements on predictive accuracy:

- Input Set A: HR + BR + IMU Torso
- Input Set B: HR + BR + IMU Torso + IMU Arm
- Input Set C: HR + BR + IMU Torso + IMU Leg
- Input Set D: HR + BR + IMU Torso + IMU Arm + IMU Leg

## Statistics

In this study, we explore the impact of different IMU data representations on model accuracy by comparing features derived from raw data (RAW) versus engineered data (MAD). We then evaluate the influence of various sensor configurations on the accuracy of $VO_2$ predictions. Subsequently, we conduct a residual analysis to assess the model's prediction ability. This involves calculating residuals as the difference between the measured $VO_2$ values and the predicted $VO_2$ values. To quantify the accuracy of the predictions, we calculate the mean absolute error (MAE) and the RMSE for the residuals.

Further, a regression analysis of these residuals yields Pearson's correlation coefficient (r) and explains the proportion of variance ($R^2$) accounted for by each model. Additionally, we employ a Bland-Altman analysis to assess the agreement between the measured and predicted VO$_2$ values, calculating both the mean bias and the limits of agreement at a 95% confidence interval (twice the standard deviation). The Bland-Altman plot is particularly suited for this study as it provides a visual representation of the agreement between two measurement methods (predicted and measured VO$_2$ values), highlighting systematic biases and variability across the range of measurements. This complements traditional error metrics like MAE and RMSE by offering insight into how prediction errors vary with VO$_2$ magnitude, thus enhancing the interpretability of the model's performance.

## Results

**Model performance across configurations:** Table 2 presents the performance metrics (RMSE and MAE) for the tested machine learning models using different input configurations and data representations. The LSTM model with RAW data and Set C input configuration achieved the best overall performance, with the lowest RMSE (4.976) and MAE (3.698 mL · kg$^{-1}$ · min$^{-1}$) on the test set. This indicates the LSTM model had the most accurate predictions among all tested configurations.

Other models, such as CNN and MLP, also demonstrated competitive results. For instance, the CNN with RAW data and Set C achieved an MAE of 4.174 mL·kg$^{-1}$·min$^{-1}$, while the MLP with the same configuration achieved an MAE of 4.326 mL · kg$^{-1}$ · min$^{-1}$. Models using MAD data generally performed less effectively than those using RAW data.

The analysis of sensor configurations revealed that multi-sensor setups, such as Set B (torso and arm sensors) and Set C (torso and leg sensors), provided the highest prediction accuracy. Single-sensor setups, such as Input A (torso only), also yielded acceptable performance but were slightly less accurate.

**Evaluation of agreement and prediction bias:** Fig 3a illustrates the linear relationship between the predicted and measured VO$_2$ values for the LSTM model using RAW data and

**Table 2. Performance metrics of ML models.**

| Model | Data | Input | Valid | | Test | |
|---|---|---|---|---|---|---|
| | | | RMSE | MAE | RMSE | MAE |
| LSTM | RAW | Set C | 4.1113 | 3.2988 | 4.9763 | 3.6979 |
| MLR | MAD | Set B | 7.2175 | 6.3824 | 5.0101 | 3.7977 |
| MLR | MAD | Set D | 7.2620 | 6.4323 | 5.0861 | 3.8283 |
| MLR | RAW | Set A | 6.2807 | 5.2638 | 4.9406 | 3.9537 |
| MLR | MAD | Set A | 7.1939 | 6.3416 | 5.0850 | 3.9756 |
| MLR | RAW | Set C | 5.7539 | 4.7465 | 5.2296 | 4.0882 |
| CNN | RAW | Set C | 4.5602 | 3.7178 | 5.6945 | 4.1747 |
| MLR | MAD | Set C | 7.2695 | 6.4395 | 5.4021 | 4.3061 |
| MLP | RAW | Set C | 4.2032 | 3.3455 | 5.6987 | 4.3268 |
| MLP | RAW | Set A | 4.1333 | 3.0814 | 5.7361 | 4.3335 |
| LSTM | RAW | Set D | 3.9768 | 3.0669 | 5.9422 | 4.3835 |
| XGBoost | RAW | Set D | 4.4614 | 3.4353 | 5.8601 | 4.5239 |
| MLP | RAW | Set D | 4.4614 | 3.4353 | 5.8601 | 4.5239 |
| CNN | RAW | Set A | 4.1312 | 3.2282 | 6.0970 | 4.5933 |
| LSTM | RAW | Set A | 4.3833 | 3.6292 | 6.5239 | 4.7223 |

**Notes:** This table shows the performance metrics of different machine learning models, including RMSE and MAE (mL·kg$^{-1}$·min$^{-1}$) for validation and test sets.

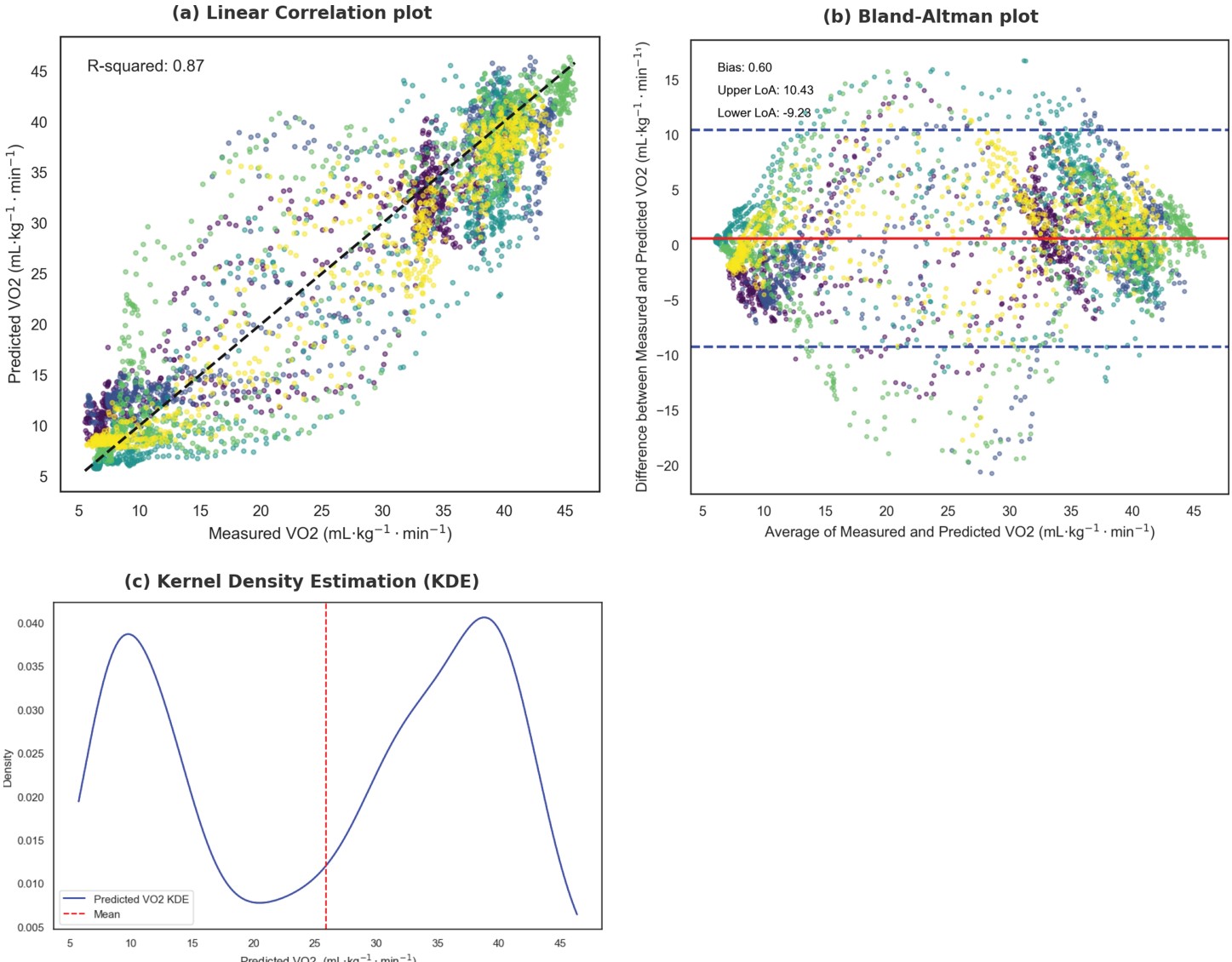

**Fig 3. (a) Linear correlation plot showing the relationship between predicted VO$_2$ and measured VO$_2$ for the LSTM model using RAW data and Set C input configuration, with an $R^2$ value of 0.87.** (b) Bland-Altman plot illustrating the difference between measured and predicted VO$_2$ values against the average of the two for all subjects combined. (c) Kernel Density Estimation (KDE) plot indicating the bimodal distribution of predicted VO$_2$ values.

the Set C configuration. The $R^2$ value of 0.87 indicates that the model accounts for 87% of the variance in the measured VO$_2$ values, showcasing strong predictive performance.

Fig 3b presents the Bland-Altman plot, which assesses the agreement between predicted and measured VO$_2$ values. The mean bias of 0.50 mL $\cdot$ kg$^{-1}$ $\cdot$ min$^{-1}$ reflects a slight tendency toward overprediction. Most data points fall within the 95% limits of agreement (Upper LoA: 10.24, Lower LoA: –9.23), indicating consistent predictions across the range of VO$_2$ values. The Bland-Altman analysis provides an additional layer of insight into prediction discrepancies, complementing conventional performance metrics.

Fig 3c shows the kernel density estimation (KDE) of the predicted VO$_2$ values, revealing a bimodal distribution with peaks in lower and higher ranges. This distribution suggests that

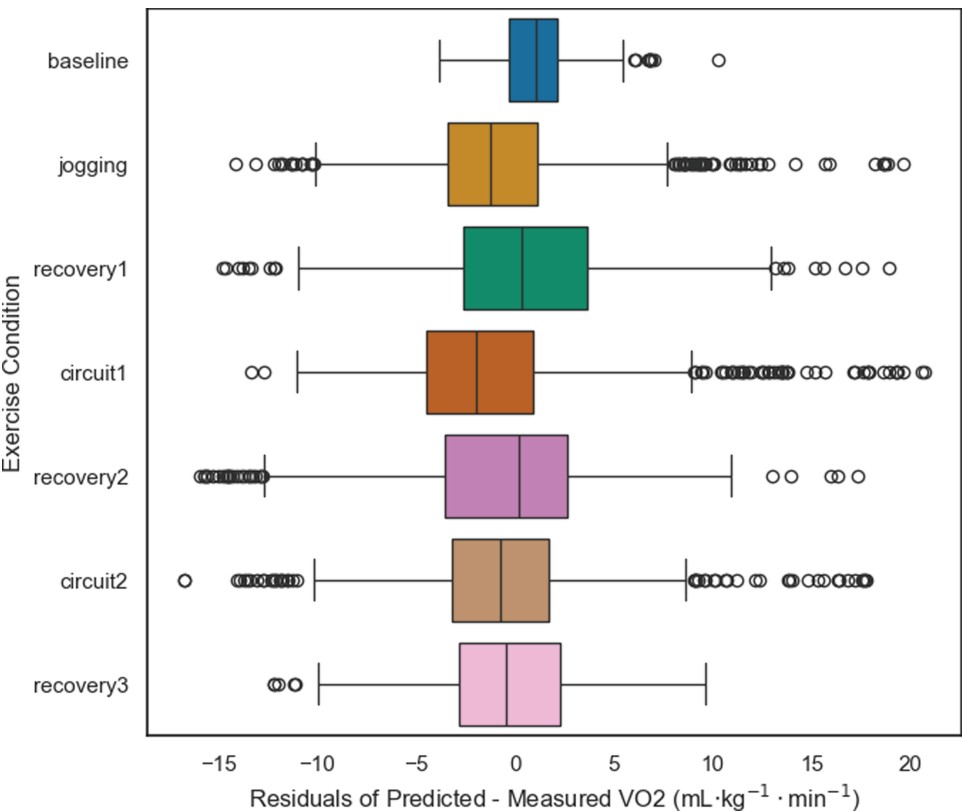

**Fig 4. The box plot illustrates the residuals (predicted VO$_2$ minus measured VO$_2$) across different exercise conditions for the LSTM model using RAW data and Set C input configuration.** The exercise conditions include baseline, jogging, recovery1, circuit1, recovery2, circuit2, and recovery3.

the model effectively differentiates between inactive and active states. However, the bimodal pattern indicates potential challenges in capturing nuanced transitions within active states or across varying intensities. This observation highlights an area for future improvements in feature engineering and model refinement to better capture intermediate states.

**Residual analysis across exercise conditions:** Residuals (predicted VO$_2$ minus measured VO$_2$) were analyzed across different exercise phases to assess the model's performance in varying conditions. Fig 4 shows that the median residual is close to zero for most conditions, indicating minimal systematic bias. However, residual variability is higher during recovery phases, suggesting challenges in accurately capturing VO$_2$ kinetics during rapid transitions between exercise and rest.

**Temporal predictions:** Fig 5 compares breath-by-breath VO$_2$ predictions with measured values for the LSTM model using RAW data and Set C. While the model tracks the overall trends of VO$_2$ changes, deviations occur during high-intensity and recovery phases. The smoothed predictions (Fig 5) reduced MAE from 3.374 to 2.902 mL $\cdot$ kg$^{-1}$ $\cdot$ min$^{-1}$, demonstrating the utility of post-processing techniques in improving predictive accuracy.

## Discussion

In this study, we investigated the ability of ML models to estimate individual VO$_2$ during simulated team sports activities using wearable sensor data. The residual analysis demonstrated

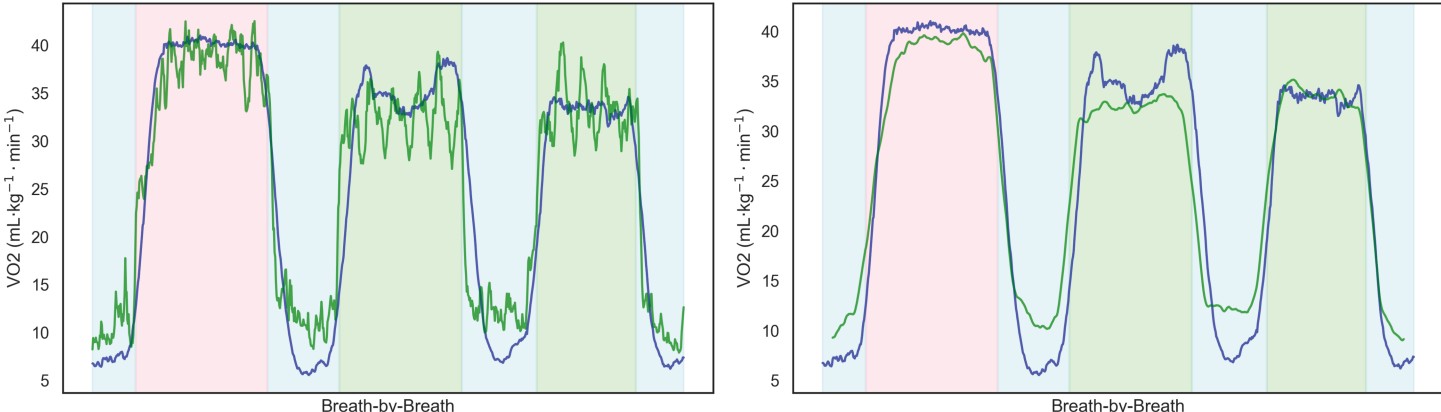

**Fig 5. The graph compares breath-by-breath VO$_2$ predictions (blue line) with measured VO$_2$ values (green line) for the LSTM model using RAW data and Set C input configuration for Subject 2.** The left plot shows unsmoothed predictions with an MAE of 3.374 (mL · kg$^{-1}$ · min$^{-1}$), while the right plot shows smoothed predictions with an MAE of 2.902 (mL · kg$^{-1}$ · min$^{-1}$). The plot includes different exercise and recovery phases, shaded as follows: baseline and recovery phases (light blue), jogging (light pink), and simulated soccer circuit (light green).

that our ML models could accurately predict measured VO$_2$ during these activities, utilising data collected from wearable sensors during fitness testing. We compare deep-learning models that have shown potential in predicting VO$_2$ kinetics during intermittent transitions [20,21,23,24] against a MLR model, which served as a baseline for comparison. Comparative analysis revealed no significant advantage of deep learning models over the baseline MLR model in terms of predictive power (Table 2). Our best MLR model achieved an MAE of 3.79 (mL · kg$^{-1}$ · min$^{-1}$), second only to the LSTM model, which achieved an MAE of 3.69 (mL · kg$^{-1}$ · min$^{-1}$) between predicted and measured VO$_2$.

We also investigated two different data representations for model performance. While deep learning models like LSTM and CNN showed strong performance with RAW data, MLR models remained competitive, particularly with MAD data representations. The choice of sensor configuration also played a significant role, with multiple sensor setups, such as torso and leg (Set C) or torso and arm (Set B), providing the most accurate predictions. A single torso-mounted sensor (Set A) notably provided good predictive performance.

This research represents the first application of ML models to predict VO$_2$ during simulated team sports activities, making direct comparisons with existing studies challenging. Some comparisons can be drawn with similar research. An LSTM model using inputs such as heart rate, mechanical power output, pedalling cadence, and respiratory frequency was employed to estimate VO$_2$ during variable high-intensity cycling exercises, achieving an MAE of approximately 3.5 (mL · kg$^{-1}$ · min$^{-1}$) and an $R^2$ value of 0.89 [24]. In our study, the LSTM model achieved an $R^2$ value of 0.87 (Fig 3a), which is closely comparable. Similar to this approach, our LSTM model was trained using data from a GXT, with two arbitrary protocols of varying intensities used to evaluate predictive performance. The same study also compared their LSTM model against two baseline analytical models, which showed $R^2$ values of 0.83 and 0.90, respectively, indicating comparable performance between the LSTM and the baseline models. Our findings align with this observation, as our baseline MLR model performed similarly to our LSTM model, achieving an $R^2$ value of 0.87. These results suggest that while deep learning models like LSTM can effectively predict VO$_2$, simpler models like MLR also offer competitive accuracy.

Comparing our performance to an LSTM model that used motion features from GNSS and IMU data during unconstrained outdoor walking and running, the reported MAE was 1.36 $(mL \cdot kg^{-1} \cdot min^{-1})$, which outperformed our best LSTM model by 2.33 $(mL \cdot kg^{-1} \cdot min^{-1})$ [23]. The experimental protocol in their study differs from ours; it involved four distinct three-minute exercise conditions, including two walking and two running sessions. These continuous conditions likely resulted in steady-state activity, which is supported by the performance of their LSTM model with HR-only input, achieving an MAE of 2.52 $(mL \cdot kg^{-1} \cdot min^{-1})$. Their best-performing LSTM model utilised 93,151 total parameters and trained for over 8,000 epochs, indicating a substantial learning capacity that could potentially lead to overfitting, particularly when working with a smaller dataset.

As the complexity of the model increases, so does the risk of overfitting [37], with only marginal improvements over simpler models. This raises the question of whether deep learning significantly benefits this particular task. Notably, the performance of our deep learning and MLR models was comparable across different data representations, with both methods featuring among the top-performing models. One possible explanation is that deep learning models may overfit the training data, as the validation results indicate Table 2. The RMSE and MAE values for the test sets were consistently higher than those for the validation sets across all models. All deep learning models, LSTM, CNN, and MLP exhibited signs of overfitting of varying degrees, with the most pronounced overfitting observed in the MLP models, followed by the CNN and LSTM models. This pattern suggests that while these models perform well on validation data, their ability to generalise to unseen test data is less robust. We employed L2 regularisation, cross-validation, and early stopping techniques to mitigate the overfitting risk. Despite these measures, the challenge of achieving robust generalisation remains, highlighting the need for further research into optimising model complexity and training strategies.

A study conducted in a simulated futsal setting demonstrated that while VO$_2$ estimation using a simple linear regression equation derived from treadmill test HR data matched measured VO$_2$ at a group level (p-value = 0.38), it failed to provide reliable predictions at an individual level. This was evidenced by weak correlations and significant bias, as indicated by a Bland-Altman analysis showing a bias of –2.8 $(mL \cdot kg^{-1} \cdot min^{-1})$, with errors reaching up to 19 $(mL \cdot kg^{-1} \cdot min^{-1})$ [38]. In contrast, our model shows a reduced bias of 0.51 $(mL \cdot kg^{-1} \cdot min^{-1})$, with limits of agreement varying by only 9 $(mL \cdot kg^{-1} \cdot min^{-1})$, demonstrating superior accuracy in reflecting individual physiological responses. Another study employed mixed-effects unpenalised linear regression model to predict VO$_2$ max using HR and accelerometer data during submaximal running. This model achieved a MAE of 2.33 $(mL \cdot kg^{-1} \cdot min^{-1})$ [39]. Our baseline MLR models have the most consistent performance of all models explored in this study, with performance for all six models presented in Table 2. These findings underscore the potential improvements that MLR models and data fusion from wearable sensors offer in enhancing VO$_2$ estimation. Linear models offer interpretability and robustness against overfitting, particularly in studies with small sample sizes [40]. Both the Silva et al. (2018) and Brabandere et al. (2018) used data from incremental fitness tests to build VO$_2$ estimation models, employing linear regression analysis of HR and VO$_2$ derived from treadmill tests, using a traditional approach [41]. Similarly, as shown in Fig 6, our incremental fitness test data demonstrated a clear linear relationship between HR and VO$_2$. However, this relationship was not as consistent during the outdoor simulated circuit test, where a significant variation in HR was observed while VO$_2$ remained steady.

One of the strengths of deep learning models is their ability to capture these transitions, but they need adequate training data to learn the patterns [20]. One of our research questions was whether we could use data from laboratory fitness tests to build ML models to predict

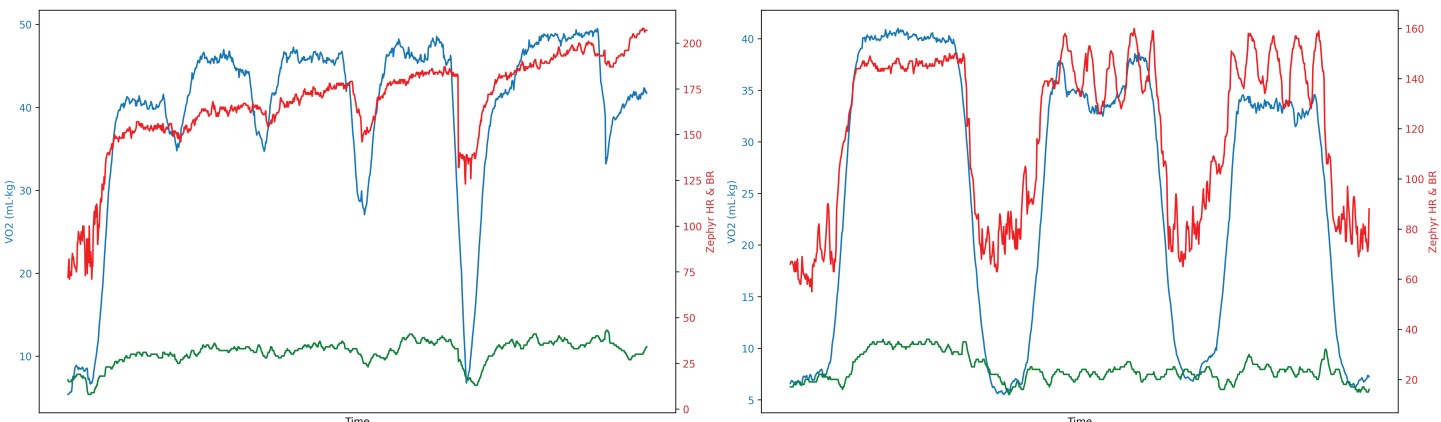

**Fig 6. Examples of oxygen uptake (VO₂) measurements (blue) and the easy-to-obtain input physiological variables are HR measurements (red) and BR (green) during the first (left) and second (right) visits for Subject 2.**

VO₂ during team sports activities. The structured exercise protocol employed during the incremental fitness test may not fully represent the unstructured activities we try to predict. Fig 4, the box plot, effectively illustrates the variability in the performance of an LSTM model for predicting VO₂ across different exercise conditions. The box plot reveals that the LSTM model with RAW data and Set C input configuration has, in general, a median residual of around 0 across different exercise conditions, indicating no significant prediction bias. However, there is noticeable variability in the residuals, particularly during the recovery phases. The model predicts more consistently during circuit activities than during the baseline and recovery phases.

Team sports' simulated activities consist of variable-speed locomotion and high-intensity actions, such as changing direction, jumping, and sprinting [13]. These activities pose unique challenges when modelling VO₂. Fig 2 shows that the raw 3D accelerometer and 3D gyroscope data can capture differences in treadmill running, outdoor running, and team-sport simulated circuits over a 3-breath window, offering high-frequency data sources to detect intense movement changes. Each circuit in our protocol included eight individual movements repeated five times, significantly increasing the number of transitions. It has been shown that IMU signals can detect high-intensity sports movements [28].

Fig 5 shows the five individual simulated circuits captured in the predicted unsmooth VO₂ output. Fig 5 shows the smoothing function applied to the input VO₂ signal is recommended for processing VO₂ during steady-state measurements. However, this smoothing operation may not be suitable for processing VO₂ data during intermittent activities. It may have removed too much dynamic information from the data for the model to learn the VO₂ kinetics, contributing to the lag in our model's performance. If transitional periods are not accounted for, this can lead to a loss of accuracy [42]. Fig 5 shows that during the recovery phase, our model struggles with these slow components, consistently overestimating the demands of recovery. When we apply the same smoothing to the VO₂ output (Fig 5) as was used on the VO₂ input, you can see underestimation during jogging and circuit 1, while circuit 2 fits well. While the model captures the general trend, significant variations and inaccuracies are observed, particularly during recovery. These discrepancies suggest that the model may require further refinement and training to improve its accuracy and reliability in tracking VO₂ dynamics during team sports activities.

One limitation of our study is the bimodal distribution observed in the predicted $VO_2$ values (Fig 3c). This pattern suggests that the model mainly distinguishes between active and inactive states. Although this behaviour aligns with physiological principles, where $VO_2$ varies significantly between rest and activity, it indicates that this distinction may dominate the predictive signal. As a result, the model may struggle to accurately capture nuanced changes within active states or during transitions between intensities. To address this, future work should consider incorporating features that could better represent transitions and intermediate states. Furthermore, expanding the dataset to include a wider range of intensities and activity transitions may improve the model's ability to fully capture $VO_2$ dynamics.

Another limitation of our study is the need for more suitable intermittent testing protocols to better represent transitional periods and the dynamic nature of team sports in the training data. Addressing this limitation is crucial for improving model accuracy and reliability, as deep learning neural networks have demonstrated the ability to predict slower $VO_2$ kinetics and transitions effectively in structured protocols [21].

Finally, our dataset consisted of six participants, reduced from an initial eight due to incomplete data, all of whom were young, healthy adult males. This limits the generalisability of our findings to a broader population. Although this sample size was sufficient to provide good predictive power for neural network models, it is widely acknowledged that larger datasets are necessary for optimal performance and generalisability in neural network approaches.

The findings of this study highlight the potential practical applications of ML models in providing personalised predictions based on individual physiological responses. These models can facilitate the assessment of an athlete's training status without the need for traditional fitness testing, offering real-time feedback that enables on-the-fly adjustments to training plans. This capability may enhance athletic performance and reduce the risk of injury by delivering precise, individualised feedback tailored to each athlete's unique physiological profile. Recent studies have shown similar benefits, where ML indices predicted soccer players' training status using heart rate and other physiological data, correlating strongly with submaximal run test outcomes [43].

## Conclusion

This pilot study demonstrates the feasibility of using ML models to predict $VO_2$ using wearable sensor data during simulated team sports activities. To enhance the generalisability and accuracy of these models, future research should focus on expanding the dataset and incorporating more varied and complex training protocols. An intermittent field test, such as the Yo-Yo Intermittent Recovery Test Level 1, could provide more relevant training data while capturing different fitness levels [44]. Once fully developed, ML models could enable non-invasive monitoring of $VO_2$ during training sessions and competitive matches. By integrating wearable sensors with advanced algorithms, these models can provide real-time, individualised feedback, optimising athlete performance and well-being.

## Author contributions

**Conceptualization:** Dermot Sheridan, Arne Jaspers, Niall M. Moyna.

**Data curation:** Dermot Sheridan.

**Formal analysis:** Dermot Sheridan, Dinh Viet Cuong.

**Investigation:** Dermot Sheridan.

**Methodology:** Dermot Sheridan, Dinh Viet Cuong, Tim Op De Beck, Niall M. Moyna.

**Project administration:** Dermot Sheridan.

**Supervision:** Niall M. Moyna, Toon T. de Beukelaar, Mark Roantree.

**Visualization:** Dermot Sheridan.

**Writing – original draft:** Dermot Sheridan.

**Writing – review & editing:** Dermot Sheridan, Arne Jaspers, Toon T. de Beukelaar, Mark Roantree.

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
