## [Decision Letter · Decision Letter 0]

28 Nov 2024

PONE-D-24-39300Estimating Oxygen Uptake in Simulated Team Sports Using Machine Learning Models and Wearable Sensor DataPLOS ONE

Dear Dr. Sheridan,

Thank you for submitting your manuscript to PLOS ONE. After careful consideration, we feel that it has merit but does not fully meet PLOS ONE’s publication criteria as it currently stands. Therefore, we invite you to submit a revised version of the manuscript that addresses the points raised during the review process.

The paper has been reviewed by two reviewers, and their opinios differ. One has raised serious concerns and recommended rejction whereas the other has raised some concerns and recommended revisions. I have reviewed it myself as well and I think authors should be given a chance to address the reviewers' concerns.

We look forward to receiving your revised manuscript.

Kind regards,

Noman Naseer, PhD

Academic Editor

PLOS ONE

Journal Requirements:

[This work was conducted with the financial support of Science Foundation Ireland under grant numbers SFI/12/RC/2289_P2 and SFI/18/ CRT/6223.

SFI, Insight Research Centre for Data Analytics, URL: https://www.sfi.ie/sfi-research-centres/insight, SFI/12/RC/2289_P2, Mark Roantree

SFI, Center for Research Training in Artificial Intelligence, URL: https://www.crt-ai.ie, SFI/18/ CRT/6223, Dermot Sheridan.].

5. In the online submission form, you indicated that [All data is available by contating the author.].

7. Please amend the manuscript submission data (via Edit Submission) to include author Dr. Niall M. Moyna.

8. Please amend either the abstract on the online submission form (via Edit Submission) or the abstract in the manuscript so that they are identical.

Additional Editor Comments:

The paper has been reviewed by two reviewers, and their opinios differ. One has raised serious concerns and recommended rejection whereas the other has raised some concerns and recommended revisions. I have reviewed it myself as well and I think authors should be given a chance to address the reviewers' concerns.

Reviewers' comments:

Reviewer's Responses to Questions

**Comments to the Author**

1. Is the manuscript technically sound, and do the data support the conclusions?

Reviewer #1: Yes

Reviewer #2: Partly

2. Has the statistical analysis been performed appropriately and rigorously? 

Reviewer #1: Yes

Reviewer #2: Yes

3. Have the authors made all data underlying the findings in their manuscript fully available?

Reviewer #1: No

Reviewer #2: No

4. Is the manuscript presented in an intelligible fashion and written in standard English?

Reviewer #1: Yes

Reviewer #2: Yes

5. Review Comments to the Author

Reviewer #1: The article is very well-written, with a proper introduction to the topic in the introduction section. I like how the abstract is in a very simplistic manner elaborates on estimation of VO2 data and refers to what the study attempts to do increase its accuracy.

The data sample of 6 people, however, is smaller to make generalized conclusions about the Machine Learning models created by the study. Moreover, the authors have systematically presented the problem they are addressing at hand to make assessment of VO2 max easier for athletes.

In the methodology, there is a need to include the results obtained from the initial protocols and tests (exercises) performed by the athletes, while they were being selected for the study. Moreover, the measurement setup needs to be explained more. Furthermore, the gyroscope and accelerometer data need to be elaborated for the readers clarity.

Within the Machine Learning section, there is a need to include evidences to support use of LOSO cross validation, and grid search technique. Moreover, the authors need justify the use of Bland-Altman analysis and regression analysis.

In the results section, the authors need to explain the performance metrics in the text. This section ends abruptly with just explaining the inaccuracies of the previous model. Presently, the discussion section contains ample newer points which could have been part of the results section. The authors need to reconfigure this and orient the sections again.

The authors mention in the conclusion section that this is a pilot study and future studies need to configure this model for a wider data set. This notion should also reflect in the title and abstract of the study.

Overall, the idea and work seem sound but I would definitely request the authors to revisit and rework on the manuscript based on the given feedback.

Reviewer #2: This papers compares several machine learning algorithms for the prediction of the VO2 in 6 athletes, using wearable IMU devices. Whereas the topic is worth investigating, the novelty seems limited, as there are already previous publications doing this prediction, as stated in the manuscript. There are also some signs that the experimental methodology might need to be improved to actually address VO2 prediction, rather than activity detection:

- The Leave-One-Subject-Out cross-validation idea is valid, but data from the first session of the test subject was used in the training set. It would be necessary to know the performance of the proposal in a real Leave-One-Subject-Out scenario, where no data from the test subject is used for training, to assess the performance in future, "unregistrated" subjects.

- The predicted value appears to follow a strongly bimodal distribution (figs 3 and 5), so the prediction is largely reduced to distinguishing between low and high values of the VO2. This, in combination with the previous comment, leads me to think that the algorithm might actually learn to distinguish between two states (active/inactive) for each athlete, and their corresponding VO2 levels, rather than actually linking movement and VO2.

- The abstract is misleading: 8 athletes started the study, but only 6 are actually included in the dataset.

- Line 152: the Zephyr data was "processed", but this processing is not described.

- PCA is used prior to linear regression to reduce the dimensionality of the data, but there appear to be only 6 features, which would make PCA largely unnecessary. Also, the number of kept dimensions is not mentioned.

- The data repository is not provided.

6. PLOS authors have the option to publish the peer review history of their article (what does this mean?). If published, this will include your full peer review and any attached files.

Reviewer #1: No

Reviewer #2: No

---

## [Author Response · Author response to Decision Letter 1]

16 Jan 2025

Reviewer 1 Comments and Responses

Comment 1: The article is very well-written, with a proper introduction to the topic

in the introduction section. I like how the abstract elaborates on the

estimation of VO2 data and refers to what the study attempts to do to

increase its accuracy.

Response: We thank the reviewer for their positive feedback on the clarity of the

introduction and abstract. We are pleased that the objectives and scope of the

study were well received.

Comment 2: The data sample of 6 people is too small to make generalised conclusions

about the Machine Learning models created by the study.

Response: We acknowledge the reviewer’s concern regarding the small sample size.

This study was designed as a pilot investigation to develop and evaluate the feasibil-

ity of using machine learning models for VO2 estimation with wearable sensors. To

address this, we have updated the manuscript to explicitly state this in the Meth-

ods section (Page 5, Lines 89–91) and the Discussion section (Page 21-22, Lines

484-489). Additionally, we have revised the title and abstract to clearly reflect the

pilot nature of the study.

Comment 3: Include results obtained from the initial protocols and tests (exercises)

performed by the athletes during selection.

Response: We appreciate the reviewer’s suggestion to include results from the ini-

tial protocols and tests performed during athlete selection. To address this, we

1

have added a summary of the baseline tests, including descriptive statistics for the

fitness parameters VO2max collected during the incremental fitness tests. This in-

formation has been incorporated into the Methods section (Page 5, Lines 89–91), To

provide additional context and detail regarding participant baseline tests, we have

included a tables to summarise the results: Table 1 (Page 5) presents individual

participant data for VO2max and resting VO2 (VOR). These additions provide a

more comprehensive view of the dataset and enhance clarity for the reader.

Comment 4: The measurement setup needs to be explained more. Furthermore, the

gyroscope and accelerometer data need to be elaborated for the reader’s

clarity.

Response: We appreciate the reviewer’s suggestion to provide additional details

about the measurement setup and the gyroscope and accelerometer data. To ad-

dress this, we have expanded the description of the measurement setup in the Meth-

ods section (Page 6-7, Lines 134–151). This includes a detailed explanation of

the placement and rationale for selecting specific body locations for the inertial

measurement units (IMUs), as well as the types of signals recorded (e.g., linear

acceleration, angular velocity, magnetic field). Additionally, we described the cal-

ibration procedure for the wearable sensors to ensure accurate and reliable data

collection. These updates provide greater clarity on the methodology and improve

transparency regarding the data acquisition process.

Comment 5: Justify the use of LOSO cross-validation and grid search technique.

Response: We appreciate the reviewer’s comment regarding the need to provide

evidence and justification for the use of LOSO cross-validation and the grid search

technique. To address this, we have expanded the Machine Learning section (Page

10, Lines 198–214) to include a detailed rationale for both methodologies. LOSO

cross-validation was chosen to mimic real-world scenarios where models must pre-

dict outcomes for unseen individuals, ensuring the model’s generalisation capability.

Additionally, we described the modified LOSO strategy used in this study, which

includes data from the first visit of the test subject during training to capture

intra-individual variability while testing on data from the second visit. We have

also elaborated on the grid search technique, explaining its role in systematically

identifying the optimal hyperparameters (e.g., number of layers and neurons) to

minimise RMSE. This ensures a robust and unbiased search across the hyperpa-

rameter space and improves model performance. These changes provide additional

clarity and context for the methodological choices made in the study.

Comment 6: Justify the use of Bland-Altman analysis and regression analysis.

Response: We appreciate the reviewer’s comment regarding the justification for the

use of Bland-Altman analysis. To address this, we have expanded the Statistics

section (Page 14, Lines 301–307) to include a detailed rationale for its inclusion.

Specifically, we highlight that Bland-Altman analysis provides a visual represen-

tation of the agreement between predicted and measured VO2 values, allowing for

the identification of systematic biases and variability across the range of measure-

ments. This approach complements traditional metrics such as MAE and RMSE

by offering insights into how prediction errors vary with VO2 magnitude, thereby

enhancing the interpretability of the model’s performance. These changes clarify

the relevance and importance of Bland-Altman analysis in the context of our study.

2

Comment 7: Explain the performance metrics in the results section. The discussion

contains points that should be part of the results section; reconfigure

and orient the sections.

Response: We appreciate the reviewer’s suggestion to improve the explanation of

performance metrics and reorganize the Results and Discussion sections. To address

this, we have made the following updates:

1. Expanded the Results section (Page 14-15, Lines 309–341) to provide detailed

explanations of performance metrics, including RMSE, MAE, and R2, with their

relevance to evaluating model performance. This ensures that the metrics are in-

terpreted clearly in the context of the study.

2. Introduced Bland-Altman analysis as a complementary evaluation method. We

included a justification for its use in assessing systematic bias and agreement be-

tween measured and predicted VO2 values, enhancing the depth of the analysis.

3. Reorganised the Results section by grouping findings into distinct themes: -

Model performance across configurations. - Evaluation of agreement and prediction

bias using Bland-Altman plots. - Residual analysis across exercise conditions. -

Temporal predictions to assess the model’s behavior in different phases.

4. Moved relevant points from the Discussion section to the Results section to

ensure all findings are cohesively presented.

These changes provide a clearer and more structured presentation of the results,

aligning with the reviewer’s feedback to improve the flow and comprehensiveness of

the manuscript.

Comment 8: The notion of this being a pilot study should also reflect in the title and

abstract.

Response: We thank the reviewer for highlighting the need to consistently convey

the pilot nature of the study throughout the manuscript. In response, we have

made the following changes:

1. Updated the title to explicitly include the term ”pilot study” to reflect the ex-

ploratory nature of this research. The revised title now reads: Estimating Oxygen

Uptake in Simulated Team Sports Using Machine Learning Models and

Wearable Sensor Data: A Pilot Study (Page 1, Line 1).

2. Revised the abstract to emphasise that this is a pilot study, explicitly mentioning

the small sample size and the exploratory intent. The abstract now includes the

statement: ”This pilot study investigates the feasibility of using machine learning

models to estimate oxygen uptake (VO2) with wearable sensors during team sports

activities.”. These revisions ensure that the pilot nature of the study is communi-

cated consistently across key sections of the manuscript, aligning with the reviewer’s

feedback.

Reviewer 2 Comments and Responses

Comment 1: The Leave-One-Subject-Out cross-validation idea is valid, but data from

the first session of the test subject were used in the training set. A

3

real Leave-One-Subject-Out scenario should exclude all test subject data

from the training set.

Response: We appreciate the reviewer’s comment regarding the LOSO cross-validation

approach. To address this, we have clarified the rationale for employing a modified

LOSO strategy:

1. The inclusion of data from the first session of the test subject in the training set

reflects a practical scenario in athlete monitoring. Preseason fitness profiles, which

are collected routinely, provide a valuable baseline for building predictive models.

Leveraging this data allows the model to account for athlete-specific fitness char-

acteristics, enabling personalised tracking and monitoring throughout the season.

This approach ensures that the model can effectively use preseason data to monitor

fitness during subsequent sessions.

2. Although a stricter LOSO approach, excluding all data from the test subject,

provides a purer generalisation test, our modified approach was deliberately chosen

to align with real-world applications in athlete monitoring systems. This modified

strategy, similar to those employed in related studies (e.g., Zignoli et al., 2020),

incorporates preseason data to improve model predictions over a season. By using

session-level data from the same individual, the model gains insight into intra-

individual variability, which is critical for practical applications.

3. This strategy strikes a balance between generalisation and utility, ensuring robust

performance while maintaining relevance to practical athlete monitoring scenarios.

We have updated the paper (Page 10, Lines 198–210) to highlight the key utility of

the modified LOSO approach in leveraging preseason fitness data, aligning it with

the practical applications of athlete monitoring systems.

Comment 2: The predicted value appears to follow a strongly bimodal distribution,

which suggests the algorithm might be distinguishing between states

(active/inactive) rather than predicting VO2.

Response: We thank the reviewer for raising this important observation about the

bimodal distribution in the predicted VO2 values. In response to this comment, we

have made the following updates:

Inclusion of Kernel Density Estimation Analysis: To address the concern

regarding bimodal distribution, we performed a Kernel Density Estimation (KDE)

analysis on the predicted VO2 values to visually assess the distribution and better

understand the model’s behavior. The KDE results are now presented in Fig-

ure 3 (sub-plot c), where the bimodal distribution is clearly illustrated. This

addition complements the previously included correlation and Bland-Altman plots

and provides a more comprehensive evaluation of the model’s predictive behaviour.

Corresponding updates have been made to the text on Page 15, Lines 340–345,

to describe this analysis and its implications.

Expanded Discussion of Bimodal Distribution as a Limitation: We have

acknowledged the observed bimodal distribution as a key limitation in our study and

included it in the Discussion section on Page 22, Lines 481–490. Specifically,

the limitation highlights that the model primarily distinguishes between active and

inactive states, potentially at the expense of capturing finer transitions within active

states or between intensities.Future work is suggested to address this limitation by

4

designing improved protocols that incorporate greater variability and transitions,

reflecting the dynamic nature of team sports.

These additions aim to provide a balanced and transparent discussion of the model’s

limitations while suggesting avenues for future improvements. We believe that these

updates effectively address the concerns of the reviewer and improve the overall

quality and depth of the manuscript.

Comment 3: The abstract is misleading: 8 athletes started the study, but only 6 are

included in the dataset.

Response: We appreciate the reviewer’s observation regarding the discrepancy in the

abstract about the number of participants included in the dataset. To address this,

we have revised the abstract to accurately reflect that the final dataset consisted

of six participants. This change ensures consistency and transparency in reporting

the study population.

Additionally, we updated the Participants section in the Methods (Page 5, Lines

89–92) to provide a detailed explanation of the reduction in the sample size. These

changes clarify the participant inclusion process and align all sections of the manuscript

with the final dataset used for analysis.

Comment 4: The Zephyr data was ”processed,” but this processing is not described.

Response: We appreciate the reviewer’s observation regarding the processing of

Zephyr data. To clarify, the Zephyr data (heart rate and breathing rate) was

recorded at a frequency of 1 Hz using the device’s built-in software. No additional

processing was applied, as the data were directly exported from the device and

merged with the breath-by-breath VO2 data collected from the Cosmed K5 system.

This merging ensured synchronisation between the datasets for subsequent analysis.

We have updated the Methods section (Page 8, Lines 168-169) to include this clari-

fication, ensuring that the process is transparently documented for reproducibility.

Comment 5: PCA is used prior to linear regression, but there appear to be only

6 features, making PCA largely unnecessary. The number of retained

dimensions is also not mentioned.

Response: We thank the reviewer for highlighting the potential redundancy of using

PCA with a small number of input features. To address this, we have clarified in the

Methods section that PCA was included to ensure consistency with preprocessing

practices and to represent the input data in an orthogonal space. While PCA was

not strictly necessary given the limited feature set, it was applied as a precautionary

measure to mitigate potential multicollinearity and align with standard workflows

in similar studies.

Additionally, we have updated the manuscript to explicitly state that all six prin-

cipal components were retained. This decision was based on experimental findings

that dimensionality reduction did not improve test scores, and retaining all com-

ponents allowed the model to fully leverage the variability in the input data. We

have included this clarification in the Methods section on Page 11, Lines 225–232.

These updates aim to enhance our methodology and address the reviewer’s concerns

effectively.

5

Comment 6: The data repository is not provided.

Response: We appreciate the reviewer’s comment regarding the availability of the

data. To address this, we have made the complete dataset publicly accessible via

Zenodo, ensuring compliance with PLOS ONE’s data-sharing policy. The dataset,

along with a detailed description and metadata, can now be accessed at the following

DOI: 10.5281/zenodo.14609092.

We have updated the Data Availability Statement in the manuscript to reflect this

change (Page 16, Lines 528–529), ensuring transparency and reproducibility of the

study.

Editorial Comments and Responses

Comment 1: Include Dr. Niall M. Moyna in the author list.

Response: We have updated the manuscript submission data to include Dr. Niall

M. Moyna as an author, ensuring that the author list is accurate and complete.

Comment 2: Ensure the abstract in the manuscript matches the one in the submission

form.

Response: We have reviewed and ensured that the abstract in the manuscript

matches the one provided in the online submission form. Both versions now ac-

curately reflect the updated participant details, key findings, and the pilot nature

of the study. This ensures consistency across all submitted materials.

Comment 3: Complete and submit the inclusivity questionnaire.

Response: We have completed PLOS’ questionnaire on inclusivity in global research

and included it as Supporting Information (File attached). This ensures compliance

with PLOS ONE’s policy to enhance transparency in reporting research conducted

outside the authors’ own country or community. Additionally, we confirm that the

manuscript reflects the principles of inclusivity and ethical research practices as

outlined in the questionnaire.

Comment 4: Ensure data and code are publicly available in a repository.

Response: In compliance with PLOS ONE’s data and code-sharing policies, w

---

## [Decision Letter · Decision Letter 1]

7 Feb 2025

Estimating Oxygen Uptake in Simulated Team Sports Using Machine Learning Models and Wearable Sensor Data: A Pilot Study

PONE-D-24-39300R1

Dear Dr. Sheridan,

We’re pleased to inform you that your manuscript has been judged scientifically suitable for publication and will be formally accepted for publication once it meets all outstanding technical requirements.

Kind regards,

Noman Naseer, PhD

Academic Editor

PLOS ONE

Additional Editor Comments (optional):

All comments have been adequetly addressed.

Reviewers' comments:

Reviewer's Responses to Questions

**Comments to the Author**

1. If the authors have adequately addressed your comments raised in a previous round of review and you feel that this manuscript is now acceptable for publication, you may indicate that here to bypass the “Comments to the Author” section, enter your conflict of interest statement in the “Confidential to Editor” section, and submit your "Accept" recommendation.

Reviewer #1: All comments have been addressed

Reviewer #3: All comments have been addressed

2. Is the manuscript technically sound, and do the data support the conclusions?

Reviewer #1: Yes

Reviewer #3: Yes

3. Has the statistical analysis been performed appropriately and rigorously? 

Reviewer #1: Yes

Reviewer #3: Yes

4. Have the authors made all data underlying the findings in their manuscript fully available?

Reviewer #1: Yes

Reviewer #3: Yes

5. Is the manuscript presented in an intelligible fashion and written in standard English?

Reviewer #1: Yes

Reviewer #3: Yes

6. Review Comments to the Author

Reviewer #1: I have reviewed the amendments done by the authors. They have carefully addressed each remark pointed out in the previous review report. I would like to appreciate their efforts in making this manuscript clear, robust and comprehensive. All the best!

Reviewer #3: All the comments and suggestions provided during the review process have been carefully addressed and incorporated into the revised manuscript. We appreciate the valuable feedback, which has helped improve the clarity and quality of the study.

As this is a pilot study, we acknowledge that future research can benefit from an expanded dataset with a larger number of subjects to further validate and generalize the findings. We look forward to building upon these results in future work by considering a broader participant pool.

7. PLOS authors have the option to publish the peer review history of their article (what does this mean?). If published, this will include your full peer review and any attached files.

Reviewer #1: No

Reviewer #3: **Yes: **Jamila Akhter

---

## [Editor Report · Acceptance letter]

PONE-D-24-39300R1

PLOS ONE

Dear Dr. Sheridan,

I'm pleased to inform you that your manuscript has been deemed suitable for publication in PLOS ONE. Congratulations! Your manuscript is now being handed over to our production team.

Kind regards,

on behalf of

Dr. Noman Naseer

Academic Editor

PLOS ONE